# The Impact of Targeted Trap–Neuter–Return Efforts in the San Francisco Bay Area

**DOI:** 10.3390/ani10112089

**Published:** 2020-11-11

**Authors:** Daniel D. Spehar, Peter J. Wolf

**Affiliations:** 1Independent Researcher, 4758 Ridge Road, #409, Cleveland, OH 44144, USA; danspehar9@gmail.com; 2Best Friends Animal Society, 5001 Angel Canyon Road, Kanab, UT 84741, USA

**Keywords:** trap–neuter–return (TNR), stray cats, feral cats, free-roaming cats, community cats, nonlethal management, population reductions

## Abstract

**Simple Summary:**

Substantial and sustained reductions in community cat populations associated with trap–neuter–return (TNR) programs have been documented in a variety of locations, including in the northeastern, midwestern, and southeastern United States, as well as Australia. The present study adds to this growing body of evidence by examining the impact of a TNR program on a population of community cats living on a two-mile section of a pedestrian trail adjacent to the San Francisco Bay. An initial population of 175 cats declined by 99.4% over the 16-year program period. Of the 258 total cats enrolled between 2004 and 2020, only one remained at the end of the program period. The results of the present study corroborate previous research findings.

**Abstract:**

Recently, a growing collection of evidence that associates trap–neuter–return (TNR) programs with substantial and sustained reductions in community cat populations across a variety of environments has emerged. Peer-reviewed studies emanating from the northeastern, midwestern, and southeastern United States, as well as Australia, document such reductions. The present study expands upon this body of evidence by examining the impact of a long-term TNR program on a population of community cats residing on a pedestrian trail adjacent to an oceanic bay located on the West Coast of the U.S. A population of 175 community cats, as determined by an initial census, living on a 2-mile section of the San Francisco Bay Trail declined by 99.4% over a 16-year period. After the conclusion of the initial count, the presence of cats was monitored as part of the TNR program’s daily feeding regimen. Of the 258 total cats enrolled in the program between 2004 and 2020, only one remained at the end of the program period. These results are consistent with those documented at the various sites of other long-term TNR programs.

## 1. Introduction

The use of trap–neuter–return (TNR) as a humane alternative to the lethal management of stray and feral cats, henceforth referred to as community cats (a term broadly used to describe unowned, free-roaming cats regardless of their level of sociability), has proliferated in the USA over the past three decades after originating in Europe in the 1950s [1,2,3]. The chief goal of TNR programs is to stop community cats from reproducing, thereby reducing their numbers over time [4]. Wolf and Hamilton [5] contend that the infeasibility of eradication, along with a lack of evidence that the common practice of intermittent culling (i.e., complaint-based shelter impoundment followed, in most cases, by lethal injection) is an effective means for reducing populations of community cats, has spurred the proliferation of TNR programs. In 2014, Boone and Slater [6] suggested that an “information vacuum” exists relative to the innumerable TNR programs carried out across the USA. They noted that up to that time, the emanation of robust data from these programs had been scarce, meaning determinations about program impacts had often been based only on anecdotal evidence [6,7,8].

Recently, a number of peer-reviewed articles have been published that begin to fill the aforementioned information void. From these studies, a growing body of evidence has emerged suggesting that TNR programs of sufficient intensity [9] are capable of producing long-term reductions in free-roaming cat populations and that with ongoing management, the reductions are sustainable for extended periods and in different environments [10,11,12,13,14]. In addition, notable declines in feline shelter intake and euthanasia have been reported from communities where programs of high-impact targeted TNR [15], shelter-based return-to-field [16], or coordinated efforts where each of these tactics are employed concurrently [17,18,19] have been implemented. One- [20] or two-year [21] studies of TNR, on the other hand, have documented population increases, similar to the short-term increases documented following the implementation of TNR in at least one long-term study [22].

The objective of the present study was to examine the impact of a TNR program on the size and dynamics of a population of community cats residing along the Pacific Coast of the United States after a 16-year observation period. Descriptive data were collected in order to assess changes in the population as well as to compare the results with those of other long-term TNR efforts. The results of the retrospective investigation of this program, which took place on a discrete section of a coastal pedestrian trail, add to the diversity of settings where TNR has been associated with long-term declines in community cat populations, including university campuses (Orlando, FL, USA [10], and Sydney, Australia [13]), an urban neighborhood (Chicago, IL, USA [11]), a private residential community (Key Largo, FL, USA [14]), and the waterfront of a small New England town (Newburyport, MA, USA [12]), and further corroborate the findings of stochastic simulation modelling [9,23]. A previously published simulation model developed from empirical data [23] was utilized by Boone et al. [9] to estimate population end points as well as preventable deaths (cats killed by lethal management plus kittens dying before adulthood) occurring over a 10-year period when various methods of free-roaming cat management (no action, removal, culling, and TNR) were tested. Both high-intensity (75% sterilization rate) and low-intensity (25% sterilization rate) scenarios were simulated for all active management methods. It was found that at a level of high intensity, “TNR offers significant advantages in terms of minimizing preventable deaths while also sustainably reducing population size” [9]. 

## 2. Materials and Methods

### 2.1. Location 

The San Francisco Bay Trail (Bay Trail) is a walking and cycling path—construction of which began in 1989 and is ongoing—that upon completion will stretch 500 miles and pass through nine counties and 47 municipalities in Northern California, USA, as it engirdles the bay after which it is named [24]. The San Francisco Bay is the largest estuary on the Pacific Coast of the USA, encompassing approximately 61 square miles (158 km^2^) [25]. Presently, the Bay Trail spans over 350 miles (~563 km) [24]. The subject TNR program encompassed a paved two-mile (3.2 km) section of the trail on the peninsula along the bay. This section of the trail is approximately five feet (1.5 meters) in width and is bordered by the rock-lined shore of the bay on one side and by a mix of roadway, grassy meadow, and marshland on the other. Various parts of the program area are surrounded by bluffs or fenced off from private property. At the request of the principals of the TNR program (as a safety measure), more specific details about the program’s location have been withheld.

### 2.2. Background 

In the early 2000s, a seemingly large yet indeterminate number of unmanaged community cats resided on or in close proximity to the aforementioned stretch of the Bay Trail. By 2004, the municipality that owns the property through which the trail passes was receiving complaints about the cats and the debris left behind by a small number of people regularly feeding the cats. At the urging of a citizen advocate, a privately funded TNR program was chosen by the city to manage the cats on the trail. The volunteer organization Project Bay Cat (PBC) was formed to carry out the trapping and returning of the cats after sterilization and vaccination as well as to manage their ongoing care. PBC operates under the umbrella of the Homeless Cat Network (HCN), a well-established local, private nonprofit TNR advocacy organization. A community education campaign was initiated to inform the public about the nascent TNR program; the city erected signs along the trail explaining TNR program protocols and publicly distributed brochures for the same purpose. Brochures were disseminated via clear weatherproof dispensers affixed to the signs that were erected on the trail by the city, given (as deemed appropriate) to trail-goers by PBC volunteers, and distributed at events by HCN. The brochures served several functions, including education of the public about TNR, to provide an easily accessible list of local resources related to pet surrender in order to discourage abandonment, and to remind the public that the program was being conducted in partnership with the city. 

The PBC program protocol called for cats to be trapped in humane box traps, transported by volunteers to a veterinary facility (most often one of two private practices that provided services to PBC at no cost or to an in-house clinic at the Peninsula Humane Society) to be sterilized, vaccinated against rabies and viral rhinotracheitis/calicivirus/panleukopenia (FVRCP), and tested for feline immunodeficiency virus (FIV) and feline leukemia virus (FeLV) before being returned to the trail. Cats enrolled in the program were ear-tipped (to indicate that they had been sterilized) and received medical care as warranted. Beginning in 2007, all cats brought to the clinic for sterilization, as well as those recaptured for medical treatment, received a microchip for future identification purposes.

An initial census of the cats residing on the trail was conducted by the founder of PBC over a three-week period in the spring of 2004. She walked the area daily (alternating between morning and evening) in order to identify locations where cats were congregating and to record the presence of individual cats. Cats were identified by their physical characteristics, (e.g., color, coat, and size) as well as by behavioral traits, such as approachability and their amenability to human touch.

After weeks of surveying the area, no new cats were being identified as part of the daily counts; thus, an initial population of 175 cats was documented. Over the next several months, feeding stations were installed as close as practicable to the locations where groups of cats were regularly observed. Because the San Francisco Bay Area is part of the Pacific Flyway for migratory birds [26], the local chapter of the Audubon Society was consulted about the placement of the feeding stations so that, as necessary, the cats could be encouraged to gather away from sensitive bird habitats and nesting areas. Over the course of the program, two of the sites where several cats were known to congregate were thought to be perilously close to sensitive bird habitat, so the feeding stations were moved away from those areas and the cats were habituated to new locations; one cat, who persisted in roaming the sensitive habitat, was relocated to a barn in a neighboring community as part of a “working cat” program. These programs, which were created to work in conjunction with TNR and/or return-to-field efforts, provide an alternative for community cats who are unable to be returned to their location of origin but who are deemed unadoptable due to temperament (i.e., low level of sociability) by allowing the cats to be relocated, most often to a place of business (e.g., brewery or garden center), where, after a period of acclimation, the cats are allowed to roam freely in order to deter the presence of rodents [27,28]. 

The maximum number of feeding stations installed across the program area was 11. Due to the configuration of the shoreline along the two-mile section of the trail that made up the program area, placement of the feeding stations occurred in three distinct zones (A–C). In Zone A, where much of the trail is flanked by either a two-lane road or marshland and the bay, feeding stations and shelters were hidden in the marsh until the rainy season when they were moved to inconspicuous spots in the rocks along the water. In Zones B and C, where most of the trail is abutted on one side by a strip of grassy meadow covered in native flora and rocky shore on the other, feeding stations and shelters were concealed amidst the foliage in the meadow. Four feeding stations were located in Zone A, one in Zone B, and six in Zone C. The feeding stations were spaced between 50 feet (~15 meters) and 0.5 miles (0.8 km) apart; placement was determined by the terrain, locations known to be frequented by most cats, and the proximity of such spots to sensitive avian habitats.

### 2.3. Data Collection

After completion of the initial census, cats were recorded and their presence tracked as they appeared at feeding stations or at adjacent locations within the program area. Descriptions of each cat’s physical characteristics and personality traits were recorded (photographs were taken as practicable). Names were assigned to most of the cats, each of whom was tracked by location; this information was recorded in notebooks and later transferred to Microsoft Word documents created for each of the feeding stations. The age category (adult or kitten ≤6 months of age, as determined by clinic staff), sex, and sterilization, vaccination, and FIV/FeLV testing status of most of the cats were recorded. Such information was not recorded for cats found dead or for those immediately adopted from the program. Additional notes describing medical treatment or other details were recorded for most of the cats. However, in some cases, such details were missing from the available records.

The feeding station census documents described above and annual summary reports provided to the municipality (extant only for 2009–2019), which included a synopsis of arrivals and departures during the previous 12 months and a review of historical population trends, were analyzed as part of the present study. Semistructured interviews were conducted with past and present leadership of PBC to provide additional context and fill information gaps (details not included in the annual reports or feeding station census documents, such as program protocols and omissions in the reported data).

### 2.4. Data Analysis

As has been performed as part of previous studies of TNR programs [10,29], descriptive statistics (e.g., percentages, mean ± SD, and median and ranges) were calculated for data pertaining to population variables (e.g., annual population totals, age, sex, mode of disposition, and time on-site). Results from the initial census in 2004 were compared to results as of June 2020 for the entirety of the program area; due to the limited availability of some data, zone-specific population comparisons were limited to the time period of 2009–2020. Available notes and the recollections of program principals (via electronic interviews) were used to make plausible estimates of enrollment dates when this precise information was absent for some cats. These estimates were deemed acceptable for the purposes of this study as descriptive statistics regarding the duration of time that select groupings of cats (based upon age category or disposition) spent in the program area are provided only to add context and to offer a more complete picture of the program; analysis of such data was not the central focus of the present study. Comparisons of results to similar TNR programs in other communities were also made; however, due to fundamental differences in program length, implementation, and environmental conditions at program locations, no statistical analysis was attempted. 

## 3. Results

### 3.1. General Findings

A total of 258 cats were recorded in the program area during the 16-year program period ending in June 2020. Approximately half (130/259) of the cats resided in Zone A, 9.7% (25/259) in Zone B, and 40.2% (104/259) in Zone C; one cat spent time living in two zones, migrating from Zone B to Zone C after 1.8 years. Overall, of cats whose sex was identified, 50.5% (107/212) were male and 49.5% (105/212) were female; the sex of 46 cats (17.8%) was undocumented. Of cats for whom an age category was assigned at enrollment, 74% (154/208) were adults and 26% (54/208) were kittens; no age category was recorded for 50 cats (19.4%). 

The average number of cats enrolled in the program per year was 15.2 (median: 3; range: 0–175); however, when the 175 cats included as part of the initial census are excluded, the average number of cats enrolled on an annual basis was 5.2 (median: 3; range: 0–27). Of the 83 cats who were enrolled after the initial census, 63% (41/65) of those for which an age category was recorded were adults and 37% were kittens (24/65); no age category was recorded for 18 of these cats.

### 3.2. Common Modes of Disposition

As of June 2020, 1 of the 258 cats enrolled in the program (<1%) remained on-site (Figure 1); 107 (41%) had been adopted; 10 (4%) were admitted into foster care; 10 (4%) were relocated to barns or other sites as part of working cat programs; 60 (23%) disappeared; 35 (14%) were euthanized due to serious illness or injury; 31 (12%) were confirmed to have died from causes other than euthanasia (e.g., struck by a vehicle and body recovered); and 4 (2%) were known to have migrated out of the program area (3 such cats were drawn to the trail for a short time in 2019—a matter of a week or two—by an unauthorized feeder, but left the program area once the unsanctioned feeding was stopped) (Table 1, Figure 2).

Cats enrolled in the program spent an average of 4.5 years on-site (median: 3.1; range: 0–15.6). Those not adopted within six months of enrollment (211), regardless of their ultimate disposition, remained an average of 5.5 years on-site (median: 4; range: 0–15.6). Of the 47 cats adopted within six months of enrollment, 42 (89.4%) were kittens.

Sterilized cats (*n* = 233) remained on-site for an average of 4.7 years (median: 3.5; range 0–15.6), whereas intact cats (*n* = 25) remained on-site for an average of 2.4 years (median: 2; range: 0–11). These findings generally correspond with those from a study conducted in Rishon LeZion, Israel, demonstrating the favorable health effects of sterilization on free-roaming cats [30].

Despite the fact that program protocol called for all cats who were trapped and brought to the clinic for sterilization to be tested for FeLV and FIV, only 1 cat (<1%) tested positive for FeLV (this cat was also symptomatic so was euthanized) and 12 cats (4.7%) tested positive for FIV. Cats who tested positive for FIV were returned to their locations of origin and monitored for health concerns (and treated as necessary); 4 were eventually adopted or relocated to a working cat program after an average of 7.5 years (median of 7.2; range, 1.6–14.1) on-site; 8 others were euthanized after residing in the program area for an average of 3 years (median of 1.2; range, 0–10.4). 

### 3.3. Program Impacts

From the start of the program in 2004 to June 2020, the total community cat population across the entirety of the program area declined by 99.4% (from 175 to 1). Between 2009 and 2020, the period for which relevant data are available, the Zone A population declined by 97.5% (from 40 to 1), and the Zone B and Zone C populations were entirely eliminated (from 5 to 0 and from 45 to 0, respectively). Ten of the 11 total feeding stations were eliminated due to the decline in the community cat population: 3 of 4 in Zone A, the single station in Zone B, and 6 of 6 in Zone C.

Overall, 233 (90.3%) of PBC’s 258 cats were sterilized (including 4 kittens who were too young or too ill prior to entering foster care to undergo surgery but who were sterilized prior to adoption and 1 cat who was already sterilized upon enrollment); these cats spent an average of 4.7 years on-site (median: 3.5; range 0–15.6). Of the 25 cats who were not sterilized, all but one were adults; after an average of 2.4 years (median: 2; range: 0–11), 44% (11) of these cats disappeared, 28% (7) were euthanized for serious illness or injury, 16% (4) died, and 12% (3) migrated out of the program area before they could be trapped.

No kittens were known to have been born in the program area after 2006. All but one of the kittens who were born in the program area, as well as those who were abandoned on or who emigrated to the trail (most often with their mothers), were eventually adopted (four after being sent to foster care); one kitten was found dead on the side of the road adjacent to the trail (Table 1).

## 4. Discussion

The purpose of the present study was to determine the impact of a 16-year TNR program on a community cat population located on a two-mile section of a pedestrian trail adjacent to the San Francisco Bay. Population size, disposition of cats from the program, and population dynamics were examined. A number of variables (e.g., program duration, program structure, available resources, climate, and differences in human and community cat population densities) make direct comparisons to other similar programs difficult. Nevertheless, notably, the results of the PBC program appear to be consistent with, or in some cases more favorable than, those recorded at sites of other long-term TNR efforts. As has occurred elsewhere [10,11,12,13,14], vigilance on the part of PBC volunteers in monitoring the program area for new arrivals (so they could be quickly enrolled) and in discouraging abandonment of cats on the trail (via consistent monitoring and as a result of public education efforts) is credited with sustaining the downward trend in cat numbers over the course of the program. 

### 4.1. Population Size

The census conducted at program inception in 2004 revealed an initial population of 175 cats residing in the program area. The 99.4% decline in population (down to a single cat) over the ensuing 16 years exceeds results from four other long-term TNR efforts: on a university campus in Orlando, FL, USA (85% reduction in population size over 28 years) [10]; on a university campus in Sydney, Australia (78% reduction in population over nine years) [13]; on the grounds of a private residential community in Key Largo, FL, USA (55% reduction over 14 years) [14]; and in an urban neighborhood in Chicago, IL, USA (54% mean reduction in colony size over 4 to 10 years and a 41% decline in total population size from 75 cats at program entry to 44 at the end of the observation period) [11] (Table 2). Results of the PBC program corroborate the findings of stochastic simulation modeling demonstrating the population reduction capabilities of TNR [9,23]. The model predicted a reduction of slightly more than 50% over 16 years based on a starting population of 200 cats and a moderate sterilization rate (i.e., 40% of the unsterilized population are sterilized every six months) [23], suggesting that PBC’s sterilization rate likely exceeded this “moderate” threshold over the 16 years reported here.

Although the available data for some of the programs shown in Table 2 are insufficient to estimate sterilization efforts at 6-month intervals, the cumulative sterilization rates documented over the duration of each program suggest that these, too, likely exceeded the “moderate” threshold described by the stochastic simulation modeling. Miller et al. [23] note: “Over a ten year period, sterilizing 40% of the naïve animals during each timestep resulted in a long-term cumulative sterilization rate of 75%.” Cumulative sterilization rates of 85%, 86%, 92%, and 100% have been documented in the aforementioned studies conducted on a university campus in Orlando, FL, USA [10]; in a private residential community in Key Largo, FL, USA [14]; in an urban Chicago, IL, USA neighborhood [11]; and on the waterfront in Newburyport, MA, USA [12], respectively.

Similarly, the percentage of the total number of cats enrolled in the program who remained on-site at the end of the observation period (<1%) was less than what was observed at the other four locations mentioned above. A review of the existing literature revealed that only the decline in population associated with a TNR program conducted on the waterfront in Newburyport, MA, USA, where an estimated population of 300 resident cats was eliminated over a 17-year period (although, detailed records were no longer extant) [12], slightly exceeded the results experienced by the PBC program (Table 2).

Feeding stations were merged or completely eliminated as the population of cats declined, as has occurred elsewhere [12,14]. By the end of the program period, all but one of the 11 feeding stations had been removed from the program area. The percentage of feeding stations eliminated exceeds what was observed in Key Largo, FL, USA (where 41 of 85 stations were eliminated over a 14-year period) [14], but is similar to what occurred in Newburyport, MA, USA (13 of 14 stations were eliminated over a 17-year span) [12] (Table 2).

### 4.2. Disposition

At the end of the program period, of the 258 total cats enrolled in the program, one remained on-site. The adoption of sociable cats and socializable kittens, some of whom first moved into foster care in preparation for adoption, contributed considerably to the observed reductions in population size and was the most common outcome for cats enrolled in the program. Adoption is widely accepted as a fundamental component of TNR best practices [4], in large part because it is effective at expediting reductions in community cat numbers when employed as a complement to sterilization [10,11,12,13,22,29,31]. Of note, when the percentage of cats adopted directly from the program (41%) is combined with the percentage that first moved into foster care (4%), the combined rate is identical to the percentage of cats adopted from the TNR program in Orlando, FL, USA (cats entering foster care were not specifically tracked as part of the Orlando program), where an 85% reduction in the number of cats took place over a 28-year period [10] (Table 2).

The percentage of cats who disappeared from the program area (23%) also closely matched what occurred in Orlando, FL, USA (24%) [10], but was somewhat lower than what took place at locations in Sydney, Australia [13], and Chicago, IL, USA [11] (Table 2). The fates of cats who disappeared are unknown; however, likely outcomes for these cats include dispersal, death by traumatic injury, and undeclared adoption. Survey data indicates that 21% of pet cats in the US are acquired directly from the “stray” population [32]; therefore, some cats who disappeared were likely taken in as pets without the knowledge of colony caregivers. In addition, it was suggested by those interviewed that the bodies of some cats who died from natural causes likely went unrecovered due to the many inconspicuous and inaccessible places along the trail for dying cats to hide (e.g., amidst tall sawgrasses or between large rocks along the shore).

The percentage of cats euthanized due to serious illness or injury (14%) and the percentage who were known to have died (12%) other than by means of euthanasia were consistent with what was observed in Sydney, Australia [13], and Key Largo, FL, USA [14], but somewhat greater than what took place in Chicago, IL, USA [11], and Orlando, FL, USA [10]. Following Swarbrick and Rand [13], we estimated a mean annual mortality rate of 5.3% for the subset of cats euthanized or known to have died (i.e., 349.5 cat-years/66 cats). This is considerably less than the estimate for cats on the University of New South Wales campus (8.1%) [13] and the estimate for pet cat mortality from a 1996 survey of US households (8.3%) [33].

### 4.3. Population Dynamics

The sustained commitment to the sterilization of community cats by PBC, beginning in 2004, resulted in a sharp decline in the number of kittens in the program area. Over the course of the 16-year program, of the cats who were assigned to an age category at enrollment (208/258), 26% (54/208) were kittens; enrollment of 79.6% (43/54) of the kittens occurred during the first two years of the program, as opposed to 20.4% (11/54) afterward (2006 to 2020). The dramatic reduction in the number of kittens enrolled in the program over time is consistent with what occurred in Orlando, FL, USA [10]. As stated above, no kittens were known to have been born in the PBC program area after 2006. A similar cessation of kitten births was observed at sites of other long-term TNR programs, where such births were eliminated after two to eight years [10,11,12,13].

The prevention of kitten births enhances the welfare of a community cat population [29] by minimizing preventable suffering (i.e., illness and injury) [34,35] and death [9] associated with unfettered feline reproduction. The absence of new kitten births in the PBC program area, along with the removal of adoptable adult cats and kittens and limiting the abandonment of pet cats, appeared to be the main contributors to the sustained decline in the community cat population over time.

## 5. Study Limitations

The limitations of the present study include those common to retrospective investigations. The absence of certain data (e.g., precise dates of enrollment, departure, and sterilization for some cats and estimated ages for cats at entry rather than simple categorizations of “kitten” or “adult”) limited the types of analysis that could be performed. The lack of access to annual summary reports (as provided by PBC to the municipality) for years 2004 through 2008 contributed to this absence of data. In addition, the PBC program was designed without control group colonies to allow for direct comparison of results. Such control groups, as established by Nutter [8], offer clear benefits to researchers but, as has been witnessed elsewhere [10,11,12,13,14], are unlikely to be integrated into efforts initiated by community-based TNR groups whose main objective is reducing population size as quickly as possible.

## 6. Conclusions

The results of the present study add to the mounting evidence indicating that TNR of sufficient duration and intensity is capable of greatly reducing a population of community cats and that, with adequate ongoing management, such results appear to be sustainable over long periods and in a variety of contexts [10,11,12,13,14]. Despite several factors that limit direct comparisons, as described above, including the ongoing lack of standardized processes for the collection of data [7,36], the similarity of results among the long-term TNR programs presented in Table 2 is notable and illustrative of this conclusion. In addition, the results presented here align with the findings of recent stochastic simulation modeling [9,23] that revealed the potential population reduction and animal welfare advantages of high-intensity TNR over alternative methods of community cat management.

## Figures and Tables

**Figure 1 animals-10-02089-f001:**
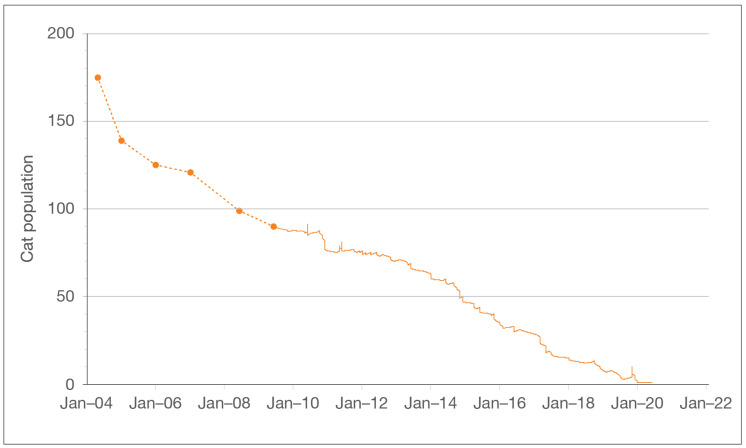
Reduction in Project Bay Cat (PBC) population over 16 years, indicating a starting population of 175 cats with one cat remaining at the end of the study period. Data prior to July 2009 (indicated by filled circles and dashed lines) was obtained from annual reports; later, more detailed data was obtained from feeding station log books.

**Figure 2 animals-10-02089-f002:**
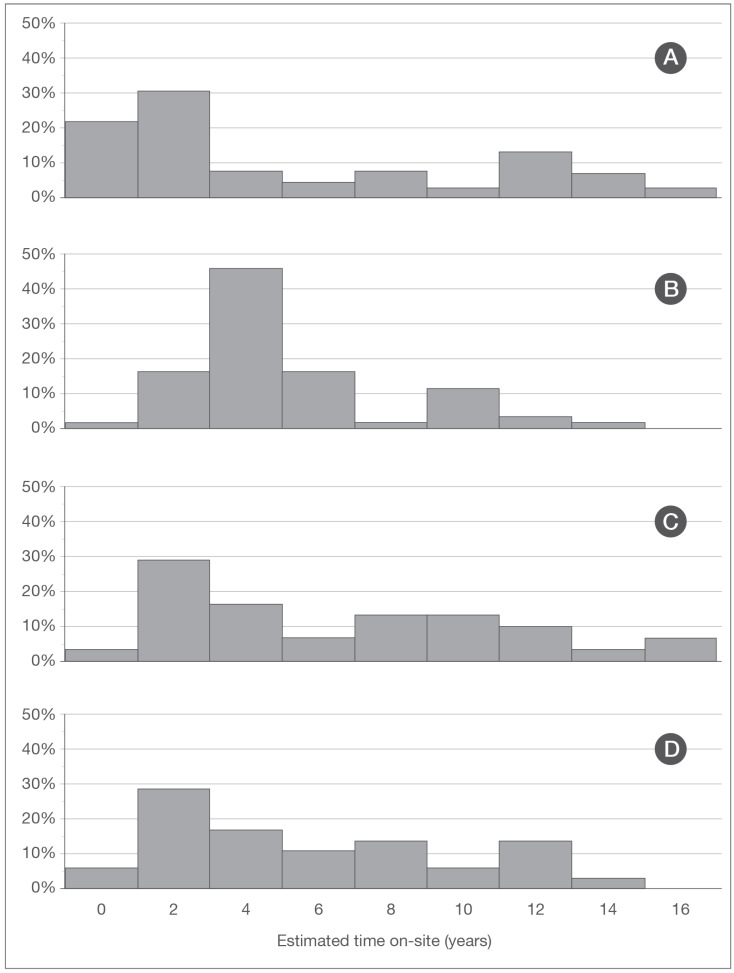
Histograms illustrating the proportion of the most common modes of disposition: adoption, foster care, and relocation ((**A**), *n* = 127); disappearance from site ((**B**), *n* = 60); death ((**C**), *n* = 31); and euthanasia ((**D**), *n* = 35).

**Table 1 animals-10-02089-t001:** Disposition of 258 community cats included in the Project Bay Cat (PBC) 16-year trap–neuter–return (TNR) program.

Final Disposition	No. of Cats by Category						Total No. of Cats (%)	Duration On-Site *		
	Sex			Age				Mean ± SD	Median	Range
	M	F	Unknown	Kitten	Adult	Unknown		(Years)	(Years)	(Years)
Remaining	1	0	0	0	1	0	1 (<1%)	9	–	–
Adopted	36	50	21	49	44	14	107 (41%)	4.3 ± 4.9	1.5	0–15.6
Foster	3	4	3	4	2	4	10 (4%)	1.7 ± 4.5	0	0–14.4
Relocated	3	6	1	0	9	1	10 (4%)	7.2 ± 4.9	6.8	0.3–12.9
Disappeared	26	23	11	0	48	12	60 (23%)	4.3 ± 2.9	3.6	0–13.6
Migrated (off-site)	1	0	3	0	0	4	4 (2%)	0.3 ± 0.3	0.1	0.1–0.8
Died	14	12	5	1	23	7	31 (12%)	5.7 ± 4.6	4.8	0–14.7
Euthanized	23	10	2	0	27	8	35 (14%)	5.0 ± 3.9	3.9	0–13.1
Total	107	105	46	54	154	50	258	4.5 ± 4.4	3.1	0–15.6

* Many of the arrival and departure dates for the cats are estimated.

**Table 2 animals-10-02089-t002:** Comparison of results from long-term trap-neuter-return (TNR) studies.

Program Location	PBC California	University of Central Florida [8]	Newburyport, MA, USA [10]	Key Largo, FL, USA [12]	Chicago, IL, USA [9]	Sydney, Australia [11]
Duration (years)	16	28	17	14	4–10	9
Cat populations						
Total managed	258	204	~340	2529	195	122
Initial census	175	68	~300	455	75 †	69
Remaining cats (no.)	1	10	0	206	44	15
(%)	1	5	0	8	23	12
Population reduction (%)	99	85	100	55	41	78
Colonies eliminated vs. total	10/11 ‡	11/16	13/14 ‡	41/85 ‡	8/20	NR
Modes of disposition						
Adoption (%)	41	45	~33	28 ^	30	27
Disappeared (%)	23	24	NR	NR	34	29
Euthanized (%)	14	11	~5–10	17 ^	3	17
Died (%)	12	8	NR	11 ^	7	12

† Total at entry for all colonies; ‡ Feeding stations; ^ Outcomes at last recorded veterinary visit; NR = not reported; PBC = Project Bay Cat.

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
