# Peer review of "The Impact of Targeted Trap–Neuter–Return Efforts in the San Francisco Bay Area"

_animals, 2020, doi:10.3390/ani10112089_

Round 1

Reviewer 1 Report

A brief summary

The study aimed to examine the effect of a trap-neuter-return (TNR) program on an existing population of cats in San Francisco Bay. Through monitoring the population for 16 years, they observed a 99.4% decline in the cat population. The authors concluded their results are similar to other TNR programs in effectiveness. The authors contribute to the growing body of literature on this topic and provide useful information on how this can be done in other areas with similar cat populations.  

Broad comments

The authors have provided a valuable contribution to the body of literature on the effectiveness of TNR programs and should be rewarded for conducting such a successful lengthy program. The manuscript is well written with minimal grammar mistakes; however, the manuscript is lacking detail in some areas and organization of presented data. I have made a number of suggestions and comments below on ways the manuscript can be improved below.

Specific comments

Simple summary

L17: Stochastic modelling introduced without sufficient context. Similar comment for the full Abstract. Suggest removing as you may not have enough space to provide sufficient explanation.

Abstract

Abstract is lacking a description of the methods; authors should provide brief details regarding how the population monitoring occurred.

Introduction

Authors should provide a clear objective at the end of this section, detailing what the precise aim of this paper is and how it will be achieved. For instance, ‘Impact of a TNR program’ is mentioned, but it would be best if it was described how this impact is being measured (e.g., population size after observation period). Hypotheses should also be stated. Authors should also mention that they aim to collect descriptive data and compare the population dynamics throughout the 16-year period as well as compare with other TNR efforts. Further, the location and background section (section 1.1 and 1.2) would benefit from a photo or graphic image of the study site with the Zones labelled on the Bay Trail.

L56: This would be an appropriate place to explain the previous research that used stochastic stimulation modelling to estimate the impact of different management actions on free-roaming cats.

L80: How were the brochures disseminated? Was the purpose of informing the public about this program so that they would not interfere?

L90-91: How were individual cats identified? Were they tagged or marked in some fashion? If a specific technique was used please explain, if not, then please describe how identification was conducted, e.g., was it based on coat colour and other obvious physical traits?

L99: Was this the only cat that was not eligible to participate? If there was specific criteria for eligibility please provide in methods.

L103: Temperament is mentioned; suggest providing an example such as low sociability to further explain this notion. If this is not what the authors mean, then revise accordingly.

L109: suggest making the labelling more distinct, “three distinct zones (labelled Zone A, B, and C)”

L112: add a “the” before trail

L117: Sensitive wildlife habitats are mentioned. Is this just referring to the bird habitats previously mentioned, or were there other habitats that you needed to consider?

Materials and Methods

L121-124: Is this how the initial census was conducted as well? Also, please provide examples as to what physical characteristics were used to identify the cats (e.g., coat colour, size, etc). What is meant by personality traits - was personality gaged in a formal assessment or was this based on whether or not the cat approached the volunteers monitoring the area, if they were always hiding, etc. Please provide objective measures as to how these factors were collected.

L125: How were the ages categorized? Why were the age of the cats not determined by the veterinary team that volunteered their services? It is mentioned elsewhere in the limitations that proper estimates of age were not attained, but it is not explained as to why this was the case. I understand the difficulty in properly aging a cat and so I am curious how a 7 month old kitten was distinguished from a 6 month old kitten. Please provide information as to how cats were categorized as kitten or adult. Further, why was 6 months of age used as a cut-off, based on AAFP Feline Life Stage Guidelines, cats are classified as junior and not ‘adult’ until around 1-2 years of age.

L130: What was detailed in the annual summary reports that was analyzed?

L133: Authors should explain what is meant by ‘information gaps’, what exactly wasn’t known that needed to be provided by the PBC leader?

L136: Please describe what data is being referred to by the term ‘population variables’.

Section 2.2 would benefit from a clear list of variables that were analyzed. Also, authors should consider reporting how the data will be presented, e.g., Percentages, mean ±SD, with median and ranges will be provided …

Results

In general, this section would benefit from subsections; First starting out with general data as the authors do (moving L189-198 to the top), then moving into a section labelled ‘Impact of program’ (or something similar), then into the common modes of disposition, and then into comparing with the other TNR programs (Table 2).

153: ‘sex of 46 cats was undocumented’. A number of cats throughout the results section did not get information recorded on them. Please explain why this was the case, either broadly or in each respective section.

L153-155: Suggest reframing this sentence to match the previous sentence for clarity. For instance, “Of the cats whose age was recorded, 74% (154/208) were adult…”.

L161: Open bracket is missing from ‘Figure1);’.  Also, it is stated that 258 cats were enrolled and the population was reduced to 1 cat, followed by a reference to Figure 1. I am not sure it is appropriate to follow this sentence with this as the figure is showing 175 cats at 2004 not 258; Reference to Figure 1 would be more appropriately placed in L183-184.

L168: I imagine these unsanctioned feeding are common in areas like this. Was this the only unsanctioned feeding that was noticed? Was there a way to control and prevent these from occurring? If the number of unsanctioned feedings are unknown, this could have influenced the number of cats enrolled or their duration of stay on the site. Authors should mention how extraneous factors such as these were controlled for, and if not, should mention how it could have influenced their gathered data.

L170: Figure legend could be more descriptive with the number of cats at the start of the program and at the end of the 16-year period.

L174: Include duration of program length in the table title.

L176-178: Figure 2: Authors should consider placing black borders around each bar, using a gray fill for the bars, and removing the border around the graph. The y-axis should also be described within the figure caption if not in the figure itself to provide more descriptive detail; Example: Displaying the proportion of cats disposed to adoption, foster care, and relocation (A, n=127)…… (D, n=35), with the respective duration of time spent on the site prior to disposal.

L187: Is there a reason why only the last remaining feeding station was kept in Zone A and not the other zones?

L195: All but one of the kittens were adopted, was this the one kitten that was found dead?

L198: Suggest removing the reference to Table 2 from this line, moving L218-228 to the results and include reference to Table 2 within that text instead. See general comments above regarding structuring the results section.

Discussion

In general, refrain from specifically referring to your tables (e.g., L225, L235, L241, L253, etc.) in the text of your discussion, this should only be done in the methods or results section. Authors should be cautious of restating the results in this section, I highlighted the times it occurred below but suggest the authors review the text for areas where specific data can be removed or re-stated in a way that is not simply repeating or stating the results.

L214-216: Please explain how your results corroborate the stochastic simulations by providing context and descriptions of these studies. This is mentioned in the abstract as well, and would help support the author’s point if they provide a clear explanation, linking to results.

L218-228: This paragraph seems redundant as it appears to just be reporting Table 2. See comment above regarding moving this section to the results. Authors should consider detailing here what specifical factors from these TNR programs they believe led to the differential success in reducing cat populations.

L243: Throughout the paper, it is either stated that 175 cats were enrolled then went down to 1, or as stated in this line, it is reported that 258 enrolled and 1 remained on site. Authors should consider consistently reporting only one starting point rather than going back and forth throughout the paper. See similar comments when discussing Figure 1.

L262: Suggest removing parentheses from the word ‘dying’ as this is specifically what is being referred to in the sentence.

L266: Citation missing following author names.

L271-272: Authors should consider rewording this sentence as it contains redundant data that is already mentioned in the results section.

L275-281: Authors should move this section to the results, as new data should not be introduced into the discussion section.

L293: Remove the ‘the’ before ‘preventable suffering’. Also, provide an example(s) as to what is meant by preventable suffering.

L294: Suggest removing “In addition to enhancing welfare” to improve clarity of the sentence.

L297: What about the monitoring procedure? Authors should consider mentioning how the monitoring techniques used and factors related to the Bay Area (e.g., people offering unsanctioned feedings) could have influenced the results of the program, either in terms of the population decline or in attaining complete data from the cat population. Further, the authors should consider making comment on the generalizability of this program to similar areas with similar cat populations and resources.

Conclusions

L313: Remove reference to Table 2.

L314: Stochastic stimulation modeling is again mentioned without sufficient background provided in the text. Authors should either consider removing this from the conclusion or incorporate this more in-depth within the discussion and/or incorporate it into the objectives as a goal to identify if the TNR program can corroborate previous findings on simulation modeling. Without doing either, the continual brief mentioning of modeling throughout the paper seems inappropriate.

Author Response

Thank you.

Reviewer 2 Report

This is a valuable short note that should be published.

The introduction is very thin.  The sections under ‘location’ and ‘background’ belong in the methods section.  This would leave 23 lines of introduction, which does not adequately establish the background/justification for such a study.  Instead, the introduction should establish

(1) the importance of recording the changes in population in response to TNR in a context that can be understood by your readers – i.e. relevance to the ultimate goals of the programs.   A key aim of TNR programs is that there is population extinguishment – which has been demonstrated in this study.

(2) the benefits of understanding the age structure of TNR populations – to my knowledge this has not been well done by many other studies and I believe that the present data could make an important contribution in this regard.  This needs additional work to extract the data from the program records, but would substantially value-add to this study.

(3) Objective description of the welfare of cats under this program. There is a blindside in the literature cited.  The cited literature does not represent a robust representation of the topic. See Calver and Fleming 2020 Animals, who have published a study of the citation bias in TNR literature – this may be informative.  For example, there is substantial discussion in the literature around the welfare of cats under TNR programs – this debate, published in Animals and which the authors have contributed to, has been ignored.

(4) There has been a strong call for identifying robust systematic methods to apply to the topic of stray animal population control methods (see Smith et al 2019, Animals, who set out a solid case for objective analysis of the literature).  Approaching the topic for population control in cats in a similar way would be beneficial. For example, not all studies show a decrease in population in response to TNR (although the literature selected for citation would suggest otherwise).  By understanding the differences in responses to various treatments, comparing TNR against alternative options, you can then test whether there is any substance to support selecting one alternative other others. 

(5) how would you rate the ‘intensity of the TNR programs’ (L43)?  If you come up with some objective measures (e.g. amount of effort expended?, proportion of initial census number neutered?, duration of program?) that could be repeated/measured across studies, then you could start to establish a protocol for comparing and contrasting between alternative methods for population control.  This would help us to advance our science – by objectively informing the most appropriate methods of management.  It requires more work, but would substantially add value to this manuscript which otherwise is lacking in substance.

Table 1 – this needs to stand independently of the text – include citations.

We support the authors’ acknowledgement of conflict of interest

Author Response

Thank you.

Reviewer 3 Report

The manuscript entitled "The Impact of Targeted Neutral-Trap Return Efforts in the San Francisco Bay Area" is a well-written study that corroborates with previous research that shows the positive impact of Trap-Neutral-Return to the field programs, producing a long term reduction in the free cat population and reduced intake of cat shelter and euthanasia. The present article is suitable for publication with minor revision.

The introduction must be rewritten, as the location and other essential information for the study must be described in the material and methods.

Line 161 add parenthesis prior to Figure 1

Did the cats receive microchip? How they were able to identify each cat during the 16 years period?

Author Response

Thank you.

Round 2

Reviewer 2 Report

I do not understand how the study can follow individual cats sufficiently to record their fate, and having assigned them to age categories, but then cannot make some estimate of survival.  The response to this is inadequate.

L47-  "From these studies, a growing body of evidence has emerged 47 suggesting that TNR programs *of sufficient intensity* [9] are capable of producing long-term reductions in free-roaming cat populations, and that with ongoing management, the reductions are sustainable for extended periods and in different environments [10–14]." 

There has some attempt to address my request: (5) how would you rate the ‘intensity of the TNR programs’?  by including sterilization rate for the present study.  Can the same data be derived from the cited literature in Table 2?  Can some attempt to address the issue raised in the introduction be made please.